# EFFICIENT HETEROGENEOUS META-LEARNING VIA CHANNEL SHUFFLING MODULATION

**Minh Hoang**
Computer Science Department
Carnegie Mellon University
Pittsburgh, PA 15213, USA
qhoang@andrew.cmu.edu

**Carl Kingsford**
Ray and Stephanie Lane
Computational Biology Department
Carnegie Mellon University
Pittsburgh, PA 15213, USA
carlk@cs.cmu.edu

## ABSTRACT

We tackle the problem of meta-learning across heterogenous tasks. This problem seeks to extract and generalize transferable meta-knowledge through learning from tasks sampled from a multi-modal task distribution. The extracted meta-knowledge can be used to create predictors for new tasks using a small number of labeled samples. Most meta-learning methods assume a homogeneous task distribution, thus limiting their generalization capacity when handling multi-modal task distributions. Recent work has shown that the generalization of meta-learning depends on the similarity of tasks in the training distribution, and this has led to many clustering approaches that aim to detect homogeneous clusters of tasks. However, these methods suffer from a significant increase in parameter complexity. To overcome this weakness, we propose a new heterogeneous meta-learning strategy that efficiently captures the multi-modality of the task distribution via modulating the routing between convolution channels in the network, instead of directly modulating the network weights. This new mechanism can be cast as a permutation learning problem. We further introduce a novel neural permutation layer based on the classical Benes routing network, which has sub-quadratic parameter complexity in the total number of channels, as compared to the quadratic complexity of the state-of-the-art Gumbel-Sinkhorn layer. We demonstrate our approach on various multi-modal meta-learning benchmarks, showing that our framework outperforms previous methods in both generalization accuracy and convergence speed.

## 1 INTRODUCTION

Few-shot learning (FSL) is a challenging problem where the goal is to learn new concepts with only a small number of labeled samples, similar to how humans learn new things by incorporating prior knowledge and context. One promising approach to tackle FSL problems is meta-learning, which learns to extract and generalize transferable meta-knowledge from a distribution of tasks and quickly adapt it to unseen tasks. Many meta-learning methods (Yoon et al., 2018; Antoniou et al., 2019; Rajeswaran et al., 2019) are built upon the model-agnostic meta learning (MAML) framework (Finn et al., 2017). The MAML method learns a single set of model parameters for any arbitrary network architecture, and then fine-tunes it to deliver high performance on unseen tasks. However, most MAML variants assume a homogeneous task distribution in which all tasks originate from the same concept domain, and assume that transferable knowledge is globally shared among all tasks (Vuorio et al., 2019). These assumptions constrain the generalization capacity of these meta-learners when handling multi-modal task distributions, for which the task-specific optimal parameters could diverge significantly from one another. For example, if the task distribution consists of modes that are far apart (e.g., animal and vehicle recognition tasks), it would be impossible to find an initialization that is simultaneously close to all modes.

Recent work has demonstrated that the generalization of MAML and, by extension, its many variants, is indeed related to the similarity of tasks in the training distribution (Zhou et al., 2021). This perspective aligns with many previous clustering approaches that aim to detect homogeneous clusters of tasks which MAML-based learners can be effectively applied (Zhou et al., 2021; Yao et al.,

2019; Vuorio et al., 2019). Zhou et al. (2021) seeks to learn an ensemble of initializations, each of which is set to represent a cluster of tasks (i.e., a mode in the task distribution). This is achieved via augmenting the MAML loss function with an assignment step. The cluster assignment heuristic, however, is conditioned on the single-mode, vanilla MAML initialization and thus is likely not optimal in a multi-modal setting.

Alternatively, Yao et al. (2019) and Vuorio et al. (2019) propose to implicitly cluster tasks using the embedding vectors of their few-shot data. In particular, Vuorio et al. (2019) applies a *modulation network* on the learned task embedding to modulate the meta-initialization of the predictor model, yielding the task-specific parameters. Yao et al. (2019) adopts a similar idea, but further imposes explicit hierarchical structure on the task space through jointly optimizing several task cluster centroids. The estimated parameter modulation is then applied to the nearest centroid based on their embedding distance. While both methods are capable of addressing the task heterogeneity challenge, they suffer from significant increase in parameter complexity since their respective modulation networks must scale with the size of the predictor model (e.g., for an average convolutional architecture with millions of parameters, the modulation network is essentially a million-output map). Even when the modulation is applied layer-wise, learning to generate that many variables is still a challenging task. This thus prevents applying these tactics on larger architectures.

To overcome this weakness, we aim to develop a new heterogeneous meta-learning strategy that efficiently captures the multi-modality of the task distribution via modulating the routing between neurons in the network, instead of directly modulating the network weights. Our approach is partially inspired by the ShuffleNet architecture (Zhang et al., 2018), which employs convolutional channel shuffling to encode a highly expressive solution space. The phenomenal success of ShuffleNet, which achieves comparable performance to state-of-the-art models that have many-fold more parameters, suggests that adapting the routing configuration (i.e., implied by the channel shuffling order) can potentially emulate the modulation of many neurons without incurring the extra computational costs.

This insight motivates us to reformulate the weight modulation network in previous heterogeneous meta-learning approaches (Yao et al., 2019; Vuorio et al., 2019) as a routing modulation network that controls task-specific shuffling of convolution channels. In particular, given a task embedding vector, our modulation network learns to generate a permutation matrix which simulates the channel shuffling operator when multiplied with the output of a convolution layer. To model this permutation network, one can adopt the Gumbel-Sinkhorn layer (Mena et al., 2018), which differentiably transforms general square matrices to discrete permutation matrices in the limit of a temperature parameter. The permutation network can be optimized via learning a mapping $f : \mathbb{R}^z \to \mathbb{R}^{C^2}$, where $z$ and $C$ are respectively the task embedding dimension and the number of convolutional channels.

However, accurately learning a dense $C \times C$ matrix given from limited training data can be challenging, especially for large convolutional networks. To overcome this, we propose an even more compact formulation of the permutation module based on the classical Beneš routing network (Beneš, 1964), which can emulate any $C$-permutation using at most $C \log_2 C$ binary switches that pairwise permute adjacent indices. Finally, to enable end-to-end learning of this compact permutation network, we approximate the discrete switches by applying the same Gumbel-softmax transformation in Mena et al. (2018) to $2 \times 2$ general matrices. The main contributions of this paper are:

1. We develop a more efficient heterogeneous meta-learning framework that estimates the different modalities in the task distribution via modulating the network routing configurations. This modulation operator takes the form of a permutation network that performs channel shuffling based on the few shot training data of a task. Our meta routing modulation (MRM) framework is presented in Section 3.2.

2. We propose a compact formulation of the above permutation network based on a continuous relaxation of the classical Beneš network (Beneš, 1964), which we call the Gumbel-Beneš layer (Section 3.3). The Gumbel-Beneš layer trains efficiently with limited data and scales better in the number of convolution channels than previous state-of-the-art baselines. Our approach is the first to draw a connection between the classical Beneš network and architecture modulation for meta learning.

3. Our framework outperforms existing methods in terms of generalization accuracy and runtime on various multi-modal meta-learning benchmarks (Section 4).

## 2 RELATED WORK

**Meta-learning.** Existing meta-learning approaches can be broadly classified into three families: metric-based, model-based, and optimization-based methods. Model-based approaches (Sukhbaatar et al., 2015; Graves et al., 2014) aim to recognize the task identity from its few-shot data and use the task identity to adjust the model state accordingly. While these methods perform well on certain task domains, they require fixing the model architecture and thus are difficult to apply on arbitrary use cases. Metric-based methods (Snell et al., 2017; Vinyals et al., 2016) learn a task similarity metric (based on observed data) which can be used to perform inference on new tasks. Sun et al. (2021) and Patacchiola et al. (2020) respectively offer a Bayesian view on metric-based and model-based meta-learning. Optimization-based methods (Finn et al., 2017; Yoon et al., 2018; Antoniou et al., 2019; Rajeswaran et al., 2019) learn a single model initialization that is amenable to fast adaption and can be applied to any arbitrary architecture. However, most existing metric-based and optimization-based methods assume that a single metric model or parameter initialization is sufficient to capture the entire task distribution.

**Heterogeneous meta-learning.** Heterogeneous meta-learning (HML) is an emerging area that develops meta-learning techniques that can generalize well to tasks drawn from a multi-modal distribution. The majority of existing HML approaches account for task heterogeneity via one of two approaches. Yao et al. (2019) and Zhou et al. (2021) explicitly maintain several local meta initializations (i.e., task clusters), to which observed tasks are assigned during training. The effectiveness of these methods depends on the quality of the many heuristics employed, such as the number of clusters and the distance metric used for cluster assignment. On the other hand, Yao et al. (2019); Vuorio et al. (2019); Liu et al. (2021); Triantafillou et al. (2021) adopt a modulation strategy that modify some components of a global meta initialization depending on some learned task embedding vector. While these approaches do not require expert understanding of the task distribution, the methods proposed by Yao et al. (2019); Vuorio et al. (2019); Liu et al. (2021) are expensive to learn, especially with large architectures, as the modulation vector scales with the number of parameters. Li et al. (2022) and Triantafillou et al. (2021) work around this scalability issue by localizing the modulation to several adapter components within the model architecture. This heuristic, however, does not modulate the remaining weights of the network, and thus assumes that the global model can adapt solely with these components. We instead a routing modulation model that can modulate the entire network at significantly more inexpensive cost.

**Routing neural networks.** Routing neural networks or neural routing refers to a technique in neural network architecture where information is selectively passed between groups of neurons based on some learned decision rules. This can be accomplished through the use of routing algorithms or specialized routing layers in a neural network. The most common form of routing is by pruning computational paths (e.g., setting certain weights to zero), which is typically used to induce sparsity (Shazeer et al., 2017) in the network for computational efficiency, or to prevent catastrophic forgetting in continual learning scenarios (Collier et al., 2020). Random channel shuffling was introduced by Zhang et al. (2018) in the context of designing compact architectures to improve model expressiveness. The ShuffleNet architecture was subsequently extended to explicitly learn the shuffling order (Lyu et al., 2020) (i.e., via optimizing for the permutation matrices that control the shuffling). Freivalds et al. (2019) proposed another differentiable neural routing formulation via a continuous approximation of the classical Beněs routing network Beneš (1964). However this approach only mimics the discrete shuffling of Beněs network in spirit, offering no guarantee that the post-shuffling information will be preserve, and thus is not suitable for channel shuffling modulation. To the best of our knowledge, neural routing for meta learning has only been considered by the work of Cai et al. (2022) in the form of heuristic pruning. In this paper, we introduce a differentiable reparameterization of the Beněs network that almost precisely models permutations, and explicitly learns to modulate channel shuffling given observed task data.

## 3 METHOD

### 3.1 PRELIMINARIES

In the meta-learning setting, we are given a task distribution $\mathcal{T}$, where each task $T_i \sim \mathcal{T}$ consists of a dataset $\mathcal{D}_i$ and a learning objective $\mathcal{L}_i$. Similar to many other meta-learning studies, we adopt a

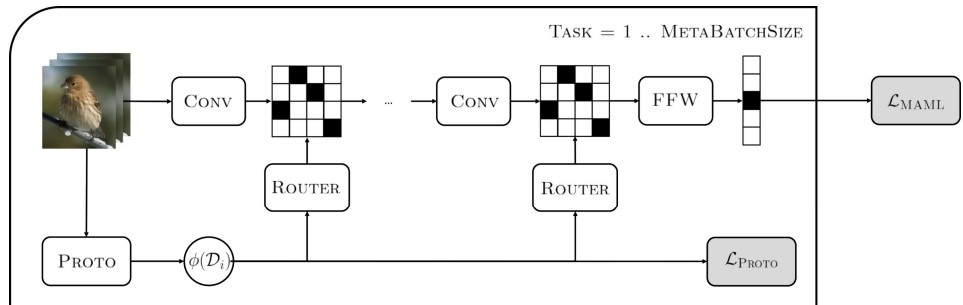

Figure 1: The overview of our meta-routing approach (MRM). Task data is first embedded as $\phi(\mathcal{D}_i)$. $\phi(\mathcal{D}_i)$ is then used to compute prototypical network loss (Snell et al., 2017) and generate channel routing matrices interleaving convolutional layers of the predictor network. Our loss function combines the MAML loss (over a meta-task batch) and the prototypical network loss (per task).

simplified setting where all tasks in $\mathcal{T}$ share the same learning objective $\mathcal{L}$ and each dataset $\mathcal{D}_i = \{\mathbf{x}_{ij}, \mathbf{y}_{ij}\}_{j=1}^n$ contains $n$-shot supervised learning samples. The goal of meta-learning is to train a meta-model $\mathcal{M}_*$ that maps any task $T_i$ to a parameter vector $\theta_i$ in the weight space of some predictor model $G$, such that $\mathcal{M}_*$ minimizes the expected learning loss (over random tasks):

$$
\begin{aligned}
\mathcal{M}_* &= \operatorname*{argmin}_{\mathcal{M}} \mathbb{E}_{T_i \sim \mathcal{T}} \left[ \frac{1}{n} \sum_{j=1}^n \mathcal{L} \left( G(\mathbf{x}_{ij}; \theta_i \triangleq \mathcal{M}(T_i)), \mathbf{y}_{ij} \right) \right] \\
&= \operatorname*{argmin}_{\mathcal{M}} \mathbb{E}_{T_i \sim \mathcal{T}} \left[ \mathcal{L}^\dagger \left( \mathcal{M}(T_i), \mathcal{D}_i) \right) \right] ,
\end{aligned}
\tag{1}
$$

where $\mathcal{L}_G^\dagger(\theta, \mathcal{D})$ denotes the averaged objective value evaluated on model architecture $G$ with parameters $\theta$ over all data points in $\mathcal{D}$. Towards this goal, the MAML framework (Finn et al., 2017) models $\mathcal{M}_*(T_i)$ as a fine-tuning gradient descent step with respect to $\mathcal{D}_i$ given some base initialization $\theta_*$. That is, $\mathcal{M}_*(T_i) \triangleq \theta_* - \eta \nabla_\theta \mathcal{L}_G^\dagger(\theta_*, \mathcal{D}_i)$, where $\eta$ denotes the step size. To obtain the base initialization $\theta_*$, Finn et al. (2017) proposes to optimize the following loss function:

$$
\theta_* = \operatorname*{argmin}_{\theta} \mathbb{E}_{T_i \sim \mathcal{T}} \left[ \mathcal{L}_G^\dagger \left( \theta - \eta \nabla_\theta \mathcal{L}_G^\dagger \left( \theta, \mathcal{D}_i^t \right), \mathcal{D}_i^v \right) \right] ,
\tag{2}
$$

where $\{\mathcal{D}_i^t, \mathcal{D}_i^v\}$ denotes the train-validation split of $\mathcal{D}_i$. Intuitively, the goal of this loss function is to find a single initialization $\theta_*$ such that, given the fine-tuning step at the time of evaluating $\mathcal{M}_*(T_i)$, the adapted parameters will yield the best performance in expectation.

## 3.2 Heterogeneous Meta Learning via Routing Modulation

Motivated by our discussion above and previous works that established that learning a single initialization $\theta_*$ is sub-optimal when the task distribution $\mathcal{T}$ is multi-modal, we now introduce our heterogeneous meta-learning approach (Fig. 1). To account for task-heterogeneity, Yao et al. (2019) and Vuorio et al. (2019) apply task-specific modulation of the base parameter $\theta_*$ as follows:

$$
\mathcal{M}_*(T_i) = \operatorname{mo}(\theta_*, \mathcal{D}_i) - \eta \nabla_\theta \mathcal{L}_G^\dagger(\operatorname{mo}(\theta_*, \mathcal{D}_i), \mathcal{D}_i) ,
\tag{3}
$$

where $\operatorname{mo}(\theta_*, \mathcal{D}_i)$ abstracts the modulation operator that takes the form $\operatorname{mo}(\theta_*, \mathcal{D}_i) = \theta_* \odot \psi(\mathcal{D}_i)$ in both Yao et al. (2019) and Vuorio et al. (2019), $\odot$ denotes the point-wise multiplication operator, and $\psi$ denotes some arbitrary embedding protocol that maps a task dataset to the weight space of the predictor $G$. For example, Vuorio et al. (2019) models $\psi$ as an attention mechanism, whereas Yao et al. (2019) pre-trains a ProtoNet task embedding (Snell et al., 2017) for task clustering and applies a fully connected network to generate $\psi(D_i)$ from the cluster centroids. Both methods, however, suffer from high additional complexity since the output dimension of $\psi$ is the prohibitively large number of parameters in $G$. To work around this shortcoming, we instead apply the task-specific modulation tactic on the architecture routing level:

$$
\mathcal{M}_*(T_i) = \theta_* - \eta \nabla_\theta \mathcal{L}_{\operatorname{mo}(G, \mathcal{D}_i)}^\dagger(\theta_*, \mathcal{D}_i) .
\tag{4}
$$

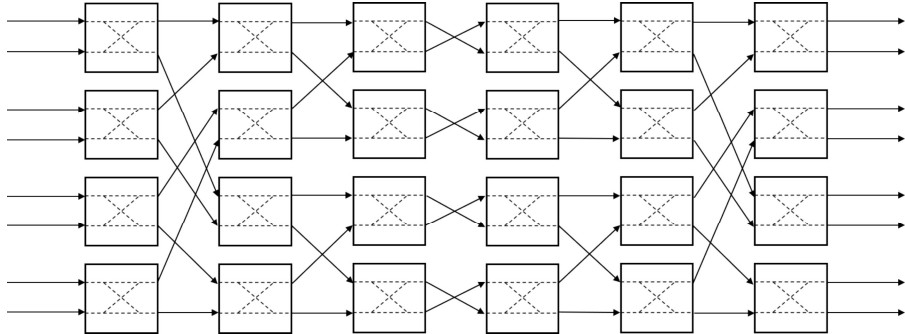

Figure 2: Beněs network for $C = 8$. Input channels are grouped pairwise. Each square block denotes a learnable binary switch that either *maintains* or *permutes* the ordering of its input pair. Routing between layers of switches are performed in a deterministic manner, where one input is routed to the top half and the other to the bottom half of the inner subnetwork (Beneš, 1964).

To concretely describe our modulation operator, we will first re-imagine the architecture $G$ as a sequence of neural layers $\{G_1, G_2, \ldots, G_M\}$, such that for any arbitrary input $\mathbf{x}$, we can rewrite $G(\mathbf{x}) = G_M \circ G_{M-1} \circ \cdots \circ G_1(\mathbf{x})$. We assume that the output of layer $G_i$ has dimension $C_i \times d_i$, where $C_i$ is the number of feature channels and $d_i$ is the (flattened) feature dimension of each channel. Then, our modulation operator can be succinctly applied through interleaving a sequence of routing layers $\{R_1, R_2, \ldots, R_{M-1}\}$ in between the predictor layers of $G$; that is:

$$\mathrm{mo}(G, \mathcal{D}_i) \quad = \quad G_M \circ R_{M-1} \circ G_{M-1} \circ \cdots \circ R_1 \circ G_1 \,, \qquad (5)$$

where each routing layer takes the form of $R_j(Q; \mathcal{D}_i) \triangleq P_j(\mathcal{D}_i)Q$ for some intermediate feature $Q \in \mathbb{R}^{C_j \times d_j}$, such that $P_j$ maps $\mathcal{D}_i$ to a $C_j \times C_j$ permutation matrix. Intuitively, the goal of $P_j$ is to re-route the information flow between layer $G_j$ and $G_{j+1}$ of the predictor net in response to the current task. It is thus appropriate for $P_j$ to generate a permutation matrix, such that the information channels are shuffled without degrading their signals.

To construct such a map, we first compute the ProtoNet embedding (Snell et al., 2017) of $\mathcal{D}_i$ and apply a convolutional layer which subsequently transforms this embedding into a general square matrix in $\mathbb{R}^{C_j \times C_j}$. To approximate the discrete permutation constraint, we could directly apply a Gumbel-Sinkhorn layer (Mena et al., 2018), whose output is guaranteed to converge to a permutation matrix in the limit of its temperature parameter. However, the Gumbel-Sinkhorn layer does not scale well with the total number of channels in $G$, and we will show in the next section that the permutation constraint can be approximated more compactly using a novel layer that we call Gumbel-Beněs. For convenience, we further let $\pi$ denote the combined weights of $\{P_1, P_2, \ldots, P_{M-1}\}$, which fully specifies our modulation network. The base predictor parameters $\theta_*$ and the modulation parameters $\gamma_*$ can be jointly optimized by extending the MAML loss function (Finn et al., 2017):

$$(\theta_*, \pi_*) = \underset{\theta, \pi}{\mathrm{argmin}} \; \mathbb{E}_{T_i} \left[ \mathcal{L}^\dagger_{\mathrm{mo}(G, \mathcal{D}_i; \pi)} \left( \theta - \eta \nabla_\theta \mathcal{L}^\dagger_{\mathrm{mo}(G, \mathcal{D}_i; \pi)} \left( \theta, \mathcal{D}^t_i \right), \mathcal{D}^v_i \right) + \lambda \mathcal{L}_{\mathrm{Proto}}(\mathcal{D}_i; \pi) \right] , (6)$$

where $\mathcal{L}_{\mathrm{Proto}}$ denotes the PROTONET loss (Snell et al., 2017) and $\lambda$ is a trade-off hyper-parameter.

### 3.3 GUMBEL-BENĚS ROUTING LAYER

The Gumbel-Sinkhorn layer is a differentiable transformation that approximately produces permutation matrices. However, in order to generate a sparse permutation matrix of size $C \times C$, it is necessary that the Gumbel-Sinkhorn layer also receives as input a dense $C \times C$ matrix. Due to this requirement, each routing layer would require at least a quadratic number of parameters in terms of $C_j$ (e.g., for the convolutional map) to generate an input to the Gumbel-Sinkhorn layer. Overall, the addition of the entire modulation component would yield an extra $\mathcal{O}(\sum_{j=1}^M C_j^2)$ learnable parameters. Although this additional complexity would be smaller than the total number of parameters in $G$ in most cases, it would become very expensive for larger architectures. To overcome this challenge, we adopt a classical result from Beneš (1964), which shows that any permutation of $C$

channels can be emulated by exactly $2\log_2 C$ layers of $C/2$ binary switches (i.e., $2 \times 2$ permutation matrix). Within each layer, we divide the input channels into groups of two and permute their features pairwise via multiplying with the content of the binary switches. The output of one layer is then forwarded to the next via a deterministic *exchange step* (Beneš, 1964) (see Appendix D for a detailed description). An example routing configuration for 8 channels is shown in Fig. 2. It was shown that the Beneš network has a congestion of 1 (Beneš, 1964), meaning there are no two different permutations that share the same switch configurations. As a result, it is sufficient to compactly model channel routing with just $C_j \log_2 C_j$ parameters at any layer $R_j$.

Finally, we can redefine our routing layer $R_j$ using the Beneš network. This is achieved by first reformulating the convolutional map to produce a stacked tensor of continuous switch configurations. We then apply the Gumbel-Sinkhorn transformation on these continuous configurations to approximate the discrete binary switches. Finally, we perform the shuffle-exchange steps recursively to permute the input channels. Formally, we describe below the computational pathway of the routing layer $R_j$ given the initial state and input $S_0 \triangleq Q$ and task data $\mathcal{D}_i$:

$$
\begin{aligned}
U_j &= f_j(\phi(\mathcal{D}_i)) , \\
\begin{bmatrix} \hat{S}_l[2k-1] \\ \hat{S}_l[2k] \end{bmatrix} &= \mathrm{GS}(U_j[l,k]) \begin{bmatrix} S_l[2k-1] \\ S_l[2k] \end{bmatrix} \quad \forall k \in [1, C_j/2] , \\
S_{l+1} &= \mathrm{exchange}\left(\hat{S}_l\right) ,
\end{aligned}
\tag{7}
$$

where $\phi$ denotes the PROTONET embedding, $f_j$ maps task embedding to its corresponding continuous switch configurations, $\mathrm{GS}(U_j[l,k])$ denotes the Gumbel-Sinkhorn transform of the component of $U_j$ corresponding to the $k^{\text{th}}$ switch of $l^{\text{th}}$ layer. The exchange step refers to the deterministic routing of the Beneš network. The entire routing system of any Beneš network for $2^n$ inputs can be efficiently constructed in a recursive manner (see Appendix D). Each $S_l$ denotes the output of the $l^{\text{th}}$ Beneš layer, and $S_{2\log_2 C}$ is the output of the routing layer $R_j$.

## 4 EXPERIMENTS

We compare the performance of our method using the Gumbel-Sinkhorn permutation layer (Mena et al., 2018) (MRM-GS) and our proposed Gumbel-Beneš routing layer (MRM-GB) against several meta-learning baselines, including MAML (Finn et al., 2017), its first-order approximation FO-MAML (Antoniou et al., 2019), prototypical network (PROTONET) (Snell et al., 2017), multimodal model-agnostic meta learning (MMAML) (Vuorio et al., 2019), universal representation learning (URL) (Li et al., 2022), and hierarchically structured meta learning (HSML) (Yao et al., 2019). We adapt the MAML and PROTONET implementations from the LEARN2LEARN package (Arnold et al., 2020). Experiments are conducted on a GTX-3080 GPU with 13GB memory. Our implementation is available at `https://github.com/Kingsford-Group/mrngb`.

For MRM-GS, MRM-GB, MMAML, HSML, we parameterize the task embedding network and the predictor network using a CNN architecture with 4 hidden convolutional blocks and a feed-forward classification layer. Each block consists of a $3 \times 3$ convolution layer, followed by BATCH-NORM, MAXPOOL and RELU activations. All convolution layers have $C = 32$ or $64$ hidden channels, depending on the specific task distribution. The mapping from task embedding to modulation parameters is parameterized by a 1-layer, TANH-activated feed-forward neural network, whose output dimension depends on the method (e.g., approximately $C^2$ for MRM-GS, $C\log_2 C$ for MRM-GB and $9C^2$ for MMAML). We apply the modulation to the first hidden convolutional layer.

MAML, FO-MAML and URL have no embedding network. For fair comparison against the above methods, we parameterize the predictor network by a two-headed CNN architecture with 4 hidden convolutional blocks per head. Their outputs are then concatenated and forwarded to the classification layer for prediction. For URL, we used the channelwise adapters suggested by Li et al. (2022) after each convolutional block as it is the most similar strategy to our channel routing layer. Last, PROTONET has no predictor network and performs prediction via clustering the input embeddings. For the same reason as above, we parameterize its embedding network by a similar two-headed CNN architectures (no classification layer).

**Meta-learning vision baselines.** The OMNIGLOT dataset (Lake et al., 2015) consists of 1623 handwritten characters from 50 different alphabets and writing systems. We randomly split the dataset

by class into train (1100 classes), validation (100 classes), and test sets (423 classes), as suggested by Ravi & Larochelle (2017). The MINI-IMAGENET dataset (Vinyals et al., 2016) is a subset of the larger ImageNet dataset (Russakovsky et al., 2015) that contains 60000 images from 100 object categories. We randomly split the dataset by category into train (64 categories), validation (16 categories), and test sets (20 categories). The JIGSAW-OMNIGLOT and JIGSAW-MINI-IMAGENET datasets are obtained by segmenting the training images in the respective original datasets into $2 \times 2$ tiles and randomly permuting these tiles to simulate 24 different task modalities. Finally, The FLOWER-AIRCRAFT-FUNGI dataset combines: (a) The VGGFLOWER102 dataset (Triantafillou et al., 2020) consisting of 102 classes of flowers (between 40 to 258 images per class); (b) the FGVCAIRCRAFT dataset (Maji et al., 2013; Triantafillou et al., 2020) consisting of 102 classes of aircraft (100 images per class); and (c) the FGVCFUNGI dataset (Triantafillou et al., 2020) consisting of 1394 classes of fungi, with a total of 89760 images.

| Methods | OMNIGLOT | MINI-IMAGENET | JIGSAW-OMNIGLOT | JIGSAW-MINI-IMAGENET |
|---|---|---|---|---|
| MAML | $0.977 \pm 0.028$ | $\mathbf{0.616 \pm 0.059}$ | $0.944 \pm 0.044$ | $0.571 \pm 0.095$ |
| FO-MAML | $0.960 \pm 0.020$ | $0.552 \pm 0.078$ | $0.921 \pm 0.068$ | $0.548 \pm 0.054$ |
| PROTONET | $0.933 \pm 0.062$ | $0.532 \pm 0.075$ | $0.860 \pm 0.054$ | $0.537 \pm 0.132$ |
| MMAML | $0.976 \pm 0.015$ | $0.604 \pm 0.146$ | $0.940 \pm 0.077$ | $0.587 \pm 0.154$ |
| HSML | $0.969 \pm 0.101$ | $0.573 \pm 0.041$ | $0.942 \pm 0.128$ | $0.572 \pm 0.096$ |
| URL | $0.971 \pm 0.048$ | $0.599 \pm 0.119$ | $0.929 \pm 0.020$ | $0.569 \pm 0.040$ |
| MRM-GS | $0.971 \pm 0.006$ | $0.613 \pm 0.151$ | $0.949 \pm 0.036$ | $0.601 \pm 0.121$ |
| MRM-GB | $\mathbf{0.981 \pm 0.011}$ | $0.614 \pm 0.005$ | $\mathbf{0.950 \pm 0.033}$ | $\mathbf{0.603 \pm 0.069}$ |

Table 1: Average test accuracy of various baseline methods on OMNIGLOT, MINI-IMAGENET datasets and their JIGSAW variants. Standard deviations are obtained over 5 random test task batches.

## 4.1 META-LEARNING FOR UNIMODAL TASK DISTRIBUTION

We show that our method performs robustly on the traditional homogeneous meta-learning setting despite the multi-modal treatment. We train all baseline methods on random batches of tasks drawn from (a) the OMNIGLOT dataset; and (b) the MINI-IMAGENET dataset. All tasks consist of randomly drawn images from 5 distinct labels. For each label, the task dataset contains $n_s$ support and $n_q$ query images. For training, both the support and query images are used to train the meta-learners. For testing, we perform fast adaptation using the support image and measure the test accuracy on the query images. We sample a batch of 32 training tasks per epoch to train each baseline method, and then evaluate their averaged performances over 5 random test tasks. We respectively let $n_s = 1$, $n_q = 15$ and $n_s = 5$, $n_q = 5$ for the OMNIGLOT and MINI-IMAGENET experiments,

Table 1 records the average test accuracy (over 5 test tasks) for each baseline method over 2000 training epochs. We defer the plots of training loss and test accuracy versus training epochs to the Appendix. Vanilla MAML achieves the best accuracy of $0.626 \pm 0.059$ on the MINI-IMAGENET dataset, and second best accuracy of $0.977 \pm 0.028$ on the OMNIGLOT dataset. These results are expected since their respective task distributions are unimodal. While other multimodal approaches (e.g., MMAML (Vuorio et al., 2019), HSML (Yao et al., 2019), and URL (Li et al., 2022)) tend to underperform in this standard setting, our methods are significantly more competitive. Specifically, on the OMNIGLOT dataset, MRM-GB achieves the best performance of $0.981 \pm 0.011$. On the MINI-IMAGENET dataset, MRM-GS and MRM-GB achieve the third and second best classification accuracies of $0.613 \pm 0.151$ and $0.615 \pm 0.005$ respectively.

## 4.2 META-LEARNING FOR MULTI-MODAL TASK DISTRIBUTION

We further conduct experiments to demonstrate the performance of our method in two different settings of task heterogeneity. In the first experiment, we simulate the multi-modality of the task distribution by applying a *jigsaw* transformation to the training images in the OMNIGLOT and MINI-IMAGENET datasets. Specifically, each training/test image is first segmented into $2 \times 2$ smaller tiles. For each sampled task, we then randomly draw a permutation of these 4 tiles and shuffle them accordingly to systematically derive new tasks that belong to $4! = 24$ different modalities.

Table 1 records the average test accuracy (over 5 test tasks) for each baseline method on the JIGSAW-OMNIGLOT and JIGSAW-MINI-IMAGENET datasets, respectively over 3000 and 5000 training epochs. In this setting, the unimodal approaches (e.g., MAML, FO-MAML, and PROTONET) generally perform worse than the multimodal approaches. This observation confirms the need for multi-modal meta learning. We further observe that PROTONET consistently demonstrates the weakest performance on these task distributions. This is most likely because PROTONET tries to assign similar embeddings to images of the same label, which include different jigsaw permutations of the same image. While our approaches also make use of the PROTONET loss to embed tasks, the specificity of the shuffling will be captured by meta-training both the predictor network and the modulator network using the MAML loss. As a result, our methods consistently achieve the best (MRM-GB) and second best (MRM-GS) classification accuracies in both datasets.

| Methods | INJECT FLOWERS | INJECT AIRCRAFT | INJECT FUNGI | INJECT ALL |
|---------|----------------|-----------------|--------------|------------|
| MAML    | $0.611 \pm 0.015$ | $0.522 \pm 0.069$ | $0.469 \pm 0.077$ | $0.491 \pm 0.115$ |
| MMAML   | $0.609 \pm 0.045$ | $0.568 \pm 0.059$ | $0.501 \pm 0.072$ | $0.527 \pm 0.049$ |
| HSML    | $\mathbf{0.615 \pm 0.007}$ | $0.540 \pm 0.101$ | $0.512 \pm 0.128$ | $0.526 \pm 0.091$ |
| URL     | $0.598 \pm 0.008$ | $0.555 \pm 0.053$ | $0.509 \pm 0.127$ | $0.512 \pm 0.041$ |
| MRM-GS  | $0.608 \pm 0.006$ | $\mathbf{0.573 \pm 0.051}$ | $0.522 \pm 0.066$ | $0.529 \pm 0.045$ |
| MRM-GB  | $0.612 \pm 0.011$ | $0.564 \pm 0.065$ | $\mathbf{0.529 \pm 0.094}$ | $\mathbf{0.537 \pm 0.063}$ |

Table 2: Average test accuracy of various baseline methods on the multi-modal FLOWER-AIRCRAFT-FUNGI dataset. We present the converged performance after injecting each modality into the task stream (at epoch $1230, 2460$ and $3690$ respectively), and the converged performance when all modalities are injected into the task stream at the beginning. Standard deviations are obtained over 5 random test task batches.

As the JIGSAW setting is naturally suited for a channel shuffling approach, we further simulate a more realistic multi-modal task distribution via grafting three different image datasets: (a) VGGFLOWERS102, (b) FGVCAIRCRAFT and (c) FGVCFUNGI (Triantafillou et al., 2020). The combined FLOWERS-AIRCRAFT-FUNGI dataset thus has three distinct task modalities. We initialize the task distribution with only data from (a) (epoch 0), and subsequently inject data from (b) after 16000 sampled train/test tasks (epoch 1230); and from (c) after 32000 sampled train/test tasks (epoch 2460s). We use a batch size of 8 tasks per epoch instead of 32 as in the above experiments.

Table 2 records the average test accuracy (over 5 test tasks) for MAML and all multi-modal meta-learning baselines. The INJECT FLOWERS, INJECT AIRCRAFT and INJECT FUNGI columns respectively report the classification performances after 1230 training epochs starting from each injection point. The INJECT ALL column reports the average test accuracy (at epoch 3690) of the meta-learner when all three modalities are introduced from the start. Additionally, Fig. 3(a) and Fig. 3(b) plot the training loss vs. epoch for each baseline on these two scenarios.

As expected, each task injection causes a degradation in terms of average test accuracy due to the introduction of a new modality in the distribution. This is also shown most clearly in Fig. 3(a), in which there is a spike in training loss following the injection of a new dataset into the task stream (i.e., marked by the vertical dotted lines). This confirms that the meta initialization learned on one modality cannot be easily adapted to address tasks from another modality, and thereby further confirms the need to address task-heterogeneity in meta-learning. Out of all baselines, MAML experiences the worst degradation, in which the classification performance at epoch 3690 is 24% worse than that at epoch 1230. Our methods MRM-GS and MRM-GB perform competitively in the unimodal setting (up to epoch 1230), and outperform other baselines after new task modalities have been injected into the task stream. We also expect that the simultaneous injection scenario is slightly easier than sequential injection. This is because the meta-learner will not be skewed towards solving the earlier introduced modalities. Indeed, we observe better performances and training losses in all baselines compared to that of the sequential injection scenario. In this setting, our method, MRM-GB still achieves the best accuracy of $0.537 \pm 0.063$.

Last, in both scenarios (i.e., *jigsaw* and *dataset grafting*), we show that our meta-routing method MRM-GB interestingly achieves an implicit clustering of tasks into the right modality. This is

demonstrated via the respective t-SNE plots of the task embeddings prior to their discrete transformation into permutation matrices (see Appendix E).

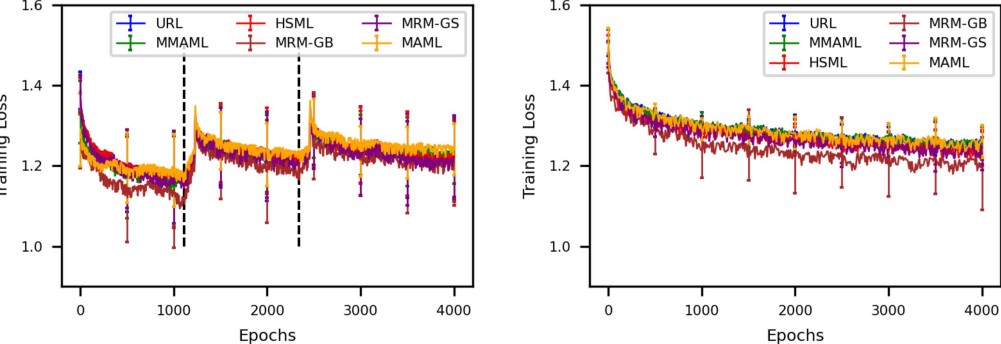

Figure 3: Training loss vs. epochs for various baseline methods on the multi-modal FLOWERS-AIRCRAFT-FUNGI dataset. We demonstrate on two settings: (a) each component dataset is introduced after 1230 training epochs in listed order (the injection points are marked by the dotted vertical lines); and (b) all component datasets are simultaneously introduced from the beginning.

### 4.3 ABLATION STUDIES

To better understand our methods, we conducted several ablation studies and defer the result to the Appendix of this paper. In particular, Appendix B shows the 2-dimensional t-SNE plot of 10000 MRN-GB learned task embeddings for two meta-learning scenarios. The t-SNE embeddings form distinct clusters separated by task modalities, and thus demonstrate that our routing method is capable of identifying related task groups. Appendix F further shows that our methods are more parameter-efficient than existing methods, especially with larger number of channels in a 4-layer convolutional architecture. Appendix G shows that our method performs robustly with the ResNet-18 architecture. Finally, we conducted ablation experiments to investigate the importance of varying the $\lambda$ parameter and the importance of routing layer placement. Our results are respectively shown in Appendix H and Appendix I.

### 5 CONCLUSION

Previous meta-learning methods assume a homogeneous task distribution, which limits their generalization ability when dealing with multi-modal task distributions. Recent works have attempted to rectify this problem, but suffer from increased complexity in terms of parameters. To overcome this limitation, we propose a novel strategy for heterogeneous meta-learning. Our approach efficiently captures the multi-modality of the task distribution by modulating the routing between convolution channels in the network. This mechanism can be viewed as a permutation learning problem, which we model and solve using a compact neural permutation layer based on the classical Benes routing network. Our Gumbel-Benes layer exhibits sub-quadratic parameter complexity in the total number of channels, in contrast to the quadratic complexity of state-of-the-art Gumbel-Sinkhorn layers. We validate our approach on various multi-modal meta-learning benchmarks, demonstrating superior performance compared to previous methods in terms of both generalization accuracy and runtime. Theoretical understanding of our approach will be a future research consideration.

**Societal Impact.** While applications of our work to real data could result in ethical considerations, this is an indirect, unpredictable side-effect of our work. Our experiment uses publicly available datasets to evaluate the performance of our algorithms; no ethical considerations are raised.

**Acknowledgements** This work was supported in part by the US National Science Foundation [DBI-1937540, III-2232121], the US National Institutes of Health [R01HG012470] and by the generosity of Eric and Wendy Schmidt by recommendation of the Schmidt Futures program. Conflict of Interest: C.K. is a co-founder of Ocean Genomics, Inc.

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
