## A    GUMBEL-BEŇES META ROUTING ALGORITHM

---

**Algorithm 1** Gumbel-Beňes Meta Routing

---

**Require:** Meta task data $\{\mathcal{D}_1, \ldots, \mathcal{D}_{n_t}\}$, convolution layers $G = \{G_1, \ldots, G_M\}$, Gumbel-Beňes routing layers $R = \{R_1, \ldots, R_M\}$ prototypical embedding network $f_{Proto}$, maximum number of epochs $n_e$, trade-off parameter $\lambda$.

  **for** $i \in [n_e]$ **do**
    $\mathcal{L} \leftarrow 0$
    **for** $t \in [n_t]$ **do**
      $\mathbf{x}_t, y_t \leftarrow \mathcal{D}_t$
      $\Phi \leftarrow f_{Proto}(\mathbf{x}_t)$ {Compute Task Embedding (see Appendix C.1}
      $\mathcal{L}_{Proto} \leftarrow \mathcal{L}_{Proto}(\mathbf{x}_t, y_t, \Phi)$ {Compute ProtoNet loss (see Appendix C.1)}
      $\mathrm{mo}(G, D_t) \leftarrow \mathbf{x}_t$
      **for** $l \in [M]$ **do**
        $\mathrm{mo}(G, D_t) \leftarrow G_l(\mathrm{mo}(G, D_t))$
        $\mathrm{mo}(G, D_t) \leftarrow R_l(\mathrm{mo}(G, D_t), \Phi)$
      **end for**
      $\mathcal{L} \leftarrow \mathcal{L} + \frac{1}{n_t} \times$ equation 6
    **end for**
    Update $G, R$ via backpropagation with loss $\mathcal{L}$
  **end for**

---

## B    TASKSET CONFIGURATION

| Tasksets | # Ways | # Shots | # Queries | # Channels |
|---|---|---|---|---|
| OMNIGLOT | 5 | 1 | 15 | 64 |
| MINI-IMAGENET | 5 | 5 | 5 | 32 |
| JIGSAW-OMNIGLOT | 5 | 1 | 15 | 64 |
| JIGSAW-MINI-IMAGENET | 5 | 5 | 5 | 32 |
| FLOWERS-AIRCRAFT-FUNGI | 4 | 1 | 5 | 16 |

Table 3: Meta task configuration for each experimental scenario. # Ways denotes the number of classes per task. (# Shots + # Queries) denotes the number of training images per class, in which # Shots and # Queries respectively denote the number of images used for the inner and outer loop update of the MAML loss. Last, # Channels denotes the number of hidden channels used per convolutional block.

## C    BACKGROUND MATERIALS

### C.1    PROTOTYPICAL NETWORK

Prototypical networks Snell et al. (2017) aim to learn an embedding function $f_{Proto}$ to encode each input into a feature vector. A class prototype of some class $c$ is the mean vector of the embedded support data samples in this class. That is:

$$\rho_c \triangleq \frac{1}{|\mathcal{S}_c|} \sum_{\mathbf{x}_i \in \mathcal{S}_c} f_{Proto}(\mathbf{x}_i; \theta) , \tag{8}$$

where $\theta$ is the parameter of $f_{Proto}$ and $\mathcal{S}_c$ is the support of class $c$. To train $f_{Proto}$, Snell et al. (2017) minimizes the negative log likelihood of the distribution induced by the prototypes:

$$
\begin{aligned}
\mathcal{L}(\theta) &\triangleq \sum_{(\mathbf{x}_i, y_i) \in \mathcal{D}} -\log P(y = y_i \mid \mathbf{x}_i) \\
&\triangleq \sum_{(\mathbf{x}_i, y_i) \in \mathcal{D}} -\log \frac{\exp \| f_{Proto}(\mathbf{x}_i) - \rho_{y_i} \|}{\sum_{c' \in \mathcal{C}} \exp \| f_{Proto}(\mathbf{x}_i) - \rho_{c'} \|} ,
\end{aligned}
\tag{9}
$$

where $\mathcal{D}$ and $\mathcal{C}$ respectively denotes the dataset and the label set. In many task representation learning frameworks, the task embedding vector is further derived from $f_{Proto}$ by concatenating all class embeddings, i.e., $\mathbf{\Phi} \triangleq [\rho_c]_{c \in \mathcal{C}}$.

## C.2 GUMBEL-SINKHORN LAYER

The Gumbel-Sinkhorn method is a technique for end-to-end learning in latent variable models that involve permutations. The Gumbel-Sinkhorn method approximates discrete permutation distribution using the continuous Sinkhorn operator, which is the analog of the softmax operator on the permutahedron. The Sinkhorn operator $S(X)$ of a general square matrix $X$ is a series of alternating row and column normalization steps of a general matrix and can be recursively written as follows:

$$
\begin{aligned}
S^0(X) &= \exp(X) \\
S^l(X) &= T_c(T_r(S^{l-1}(X))) \\
S(X) &= \lim_{l \to \infty} S^l(X) \,,
\end{aligned}
\tag{10}
$$

where $T_c$ and $T_r$ are the column-wise and row-wise normalization operators respectively. In the limit of $l$, $S(X)$ is a doubly stochastic matrix that induces a distribution of permutations (i.e., the Gumbel-Matching distribution), from which random permutations can be sampled from. To make this sampling differentiable (i.e., for the purpose of end-to-end optimization), the Gumbel-Sinkhorn method further applies the Gumbel reparameterization trick from Jang et al. (2017):

$$
\text{GS}(X) \triangleq S((X + \epsilon)/\tau) \,,
\tag{11}
$$

where $\epsilon$ is standard Gumbel noise Jang et al. (2017) and $\text{GS}(X)$ converges almost surely to samples of the Gumbel-Matching distribution as the temperature parameter $\tau$ anneals to 0.

## D BENĚS NETWORK ROUTING ALGORITHM

The Beněs network (Beněs, 1964) is constructed in a recursive manner. For the base case, the Beněs network for permuting 2 items is simply a binary switch that either maintains the order between its inputs, or reverses it. The Beněs networks for $2^n$ items with any $n > 1$ are subsequently defined using the following steps:

- **Concatenation.** We first vertically concatenate two Beněs networks for $2^{n-1}$ items to create the upper and lower sub-networks. In addition to these sub-networks, the $2^n$ Beněs network also consists of (a) an *input layer*: $2^{n-1}$ binary switches routing inputs into these sub-networks; and (b) an *output layer*: another $2^{n-1}$ binary switches routing outputs out of these sub-networks.

- **Input layer routing.** The upper output of the $k^{\text{th}}$ *input* switch is routed to the $k^{\text{th}}$ input of the upper sub-network. The lower output of the $k^{\text{th}}$ *input* switch is routed to the $k^{\text{th}}$ input of the lower sub-network.

- **Output layer routing.** The $k^{\text{th}}$ output of the upper sub-network is routed to the upper input of the $k^{\text{th}}$ *output* switch. The $k^{\text{th}}$ output of the lower sub-network is routed to the lower input of the $k^{\text{th}}$ *output* switch.

## E VISUALIZING THE TASK CLUSTERING EFFECT OF MRN-GB

We conduct a qualitative ablation study to investigate the routing quality of our method, MRN-GB. In particular, we used the MRN-GB models respectively trained on the FLOWERS-AIRCRAFT-FUNGI dataset (in the sequential setting) and the JIGSAW-OMNIGLOT dataset to generate the task embedding vectors for 10000 randomly sampled test tasks in each scenario. For visualization purpose, we use the continuous embedding vectors obtained prior to the Gumbel-Beněs transformation into the routing permutation matrices. We subsequently embed these vectors into a 2D-latent space using the t-SNE method (Van der Maaten & Hinton, 2008) and produce the resulting scatter plots in Fig. 4(a) and Fig. 4(b). The color labels in these scatter plots correspond to the ground truth

modalities of the sampled tasks (e.g., 3 modalities for each component dataset in the FLOWERS-AIRCRAFT-FUNGI scenario, and $(2 \times 2)! = 24$ modalities for each jigsaw configuration in the JIGSAW-OMNIGLOT scenario). As expected, we observe visually distinctive clusters in each scenario, which suggests that our method has successfully learned to identify and route similar tasks through similar computational pathways.

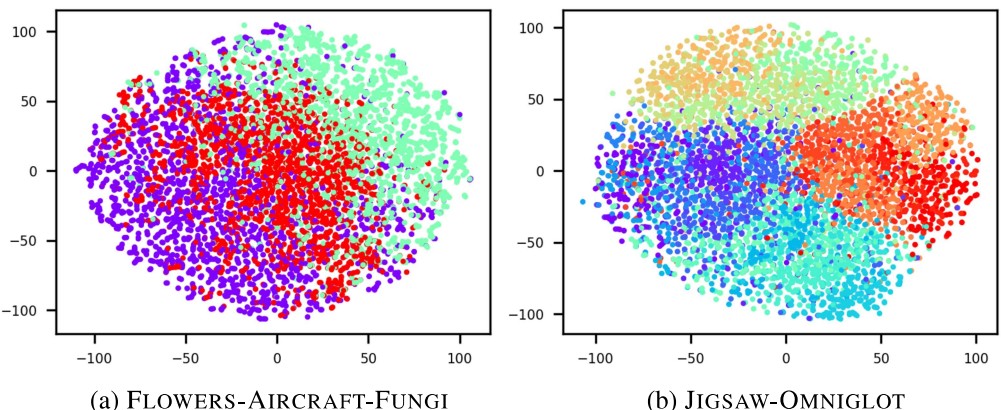

(a) FLOWERS-AIRCRAFT-FUNGI        (b) JIGSAW-OMNIGLOT

Figure 4: 2-dimensional t-SNE plots of 10000 MRN-GB task embedding vectors, where the tasks are randomly sampled from the (a) FLOWERS-AIRCRAFT-FUNGI (3-modalities) and (b) JIGSAW-OMNIGLOT datasets (24-modalities). Each color on the plot represents one modality group.

## F  SCALABILITY OF MODULATION NETWORKS

We show that our method scales well in terms of the predictor network's architecture complexity. Specifically, we vary the number of hidden channels $C$ per convolutional block in our standard 4-block CNN architecture described above and record the training time per epoch as well as the number of modulating parameters in Table 4. For this experiment, weight modulation (MMAML, HSML) and routing modulation (MRM-GB, MRM-GS) are applied to all 4 blocks. For URL, we again adopt the channelwise adapter strategy as described in Li et al. (2022). We use 4 channelwise adapters after each convolutional block for the URL model. The reported runtimes are averaged over 200 consecutive training epochs.

As expected, the number of parameters in MRM-GB and URL increase almost linearly in $C$, resulting in the most efficient runtime at every size of the predictor architecture. We note that the increase in runtime, however, does not always correlate perfectly to the increase in the number of modulation parameters. This is due to various other factors, such as the overhead cost of computing the more complicated gradients of MRM-GB, MRM-GS compared to MMAML. Nonetheless, as $C$ becomes very large, we observe that the number of modulation parameters in MRM-GS, MMAML and HSML will grow prohibitively expensive. For example, at $C = 256$ (which is reasonable for many common architectures), we were no longer able to fit the MMAML and HSML models into the GPU memory. While the complexity of URL and our MRM-GB method are on the same scale, we have shown above that our method achieves much better classification accuracy than URL. This result suggests that URL requires significantly more adapter parameters to sufficiently encode transferable knowledge, and that our method is more parameter efficient for meta learning.

## G  INVESTIGATING PERFORMANCE WITH RESNET-18 ARCHITECTURE

We further perform meta-learning on the more difficult 1-shot, 5-way MINI-IMAGENET setting and its JIGSAW counterpart. We adopt similar training settings to the 5-shot, 5-way experiments in the main manuscript. That is, we trained every baseline for 2000 epochs, sampling 32 training tasks of 16 images per epoch (1-shot, 15-queries). For testing, we sampled 5 test tasks with 1 training and 15 test images per task. Here, we only focus on comparing with other heterogeneous meta learning

| $C$ | MRM-GB | | MRM-GS | | MMAML | | HSML | | URL | |
|---|---|---|---|---|---|---|---|---|---|---|
| | $t$ | $p$ | $t$ | $p$ | $t$ | $p$ | $t$ | $p$ | $t$ | $p$ |
| 16 | 3.25 | 0.04 | 3.35 | 0.08 | 3.83 | 0.75 | 4.99 | 1.12 | 3.22 | 0.04 |
| 32 | 3.57 | 0.21 | 4.48 | 0.66 | 6.36 | 5.93 | 7.12 | 8.89 | 3.51 | 0.20 |
| 64 | 4.21 | 0.99 | 5.58 | 5.26 | 7.96 | 47.3 | 10.24 | 70.1 | 4.30 | 1.01 |
| 128 | 5.06 | 4.59 | 7.16 | 42.0 | 12.91 | 378 | 16.17 | 559 | 5.21 | 5.08 |
| 256 | 7.84 | 21.0 | 12.64 | 336 | — | 3020 | — | 4263 | 7.92 | 21.4 |

Table 4: Runtime $t$ (in seconds) per training epoch and number of learnable parameters $p$ (in millions) of various multi-modal meta-learners at different numbers of hidden convolution channels. — indicates that the model exceeds our GPU memory, and its runtime cannot be recorded.

baselines (i.e., MMAML, HSMAML, URL) as the poor compatibility of uni-modal methods have been well explored. On this same setting, we further provided the comparative performance using the architecture ResNet-18, showing that our method is robust across a wide range of methods.

| Method | MRN-GB | MRN-GS | MMAML | HSML | URL |
|---|---|---|---|---|---|
| MINI-IMAGENET (Conv) | 0.504 | 0.496 | 0.432 | 0.472 | 0.496 |
| JIGSAW MINI-IMAGENET (Conv) | 0.480 | 0.464 | 0.432 | 0.408 | 0.448 |
| MINI-IMAGENET (ResNet) | 0.547 | 0.526 | 0.541 | 0.555 | 0.470 |
| JIGSAW MINI-IMAGENET (ResNet) | 0.525 | 0.299 | 0.490 | 0.448 | 0.501 |

Table 5: Classification accuracy of various methods on the 1-shot 5-way MINI-IMAGENET setting and its JIGSAW counterpart with different architectures.

## H    IMPORTANCE OF THE TRADE-OFF PARAMETER IN THE LOSS FUNCTION

Table 6 provides the results of varying the trade-pff parameter $\lambda$ in the MRN-GB and MRN-GS objective function. We used the 1-shot 5-way Omniglot dataset and 200 training epochs for each experiment. We observe that the best performance of MRN-GB is observed with $\lambda = 1.0$, whereas $\lambda = 2.0$ is best for MRN-GS. This result seems to suggest that the parameter $\lambda$ cannot be too large, causing the loss function to behave similarly to the ProtoNet loss; or too small, causing the task embedding module to capture unmeaningful patterns.

| $\lambda$ | 0.2 | 0.5 | 1.0 | 2.0 | 5.0 |
|---|---|---|---|---|---|
| MRN-GB | 0.827 | 0.829 | 0.888 | 0.861 | 0.840 |
| MRN-GS | 0.843 | 0.848 | 0.861 | 0.885 | 0.867 |

Table 6: Classification accuracy of MRN-GB and MRN-GS on the 1-shot 5-way OMNIGLOT dataset with varying values of $\lambda$.

## I    IMPORTANCE OF ROUTER DEPTH PLACEMENT

In this section, we further investigate the impact of placing the routing layer at various depths of the network. In all of our previous experiments, we used one routing layer after every convolution block, hence the routers are distributed at various depths of the network.

Table 7 provides the result of this experiment in the 5-way 1-shot OMNIGLOT setting, in which we compared the performance of our methods using all 4 routers vs. the performance of using only one router at different depths (e.g., larger depth means being closer to the output layer). Our result suggests that having the routing layer placed earlier on in the architecture is the most effective practice (e.g., putting a routing layer after the second convolution block gives the best performance).

| Router depth | 1 | 2 | 3 | 4 | All |
|---|---|---|---|---|---|
| MRN-GB | 0.873 | 0.875 | 0.869 | 0.747 | 0.888 |
| MRN-GS | 0.851 | 0.867 | 0.835 | 0.805 | 0.861 |

Table 7: Classification accuracy of MRN-GB and MRN-GS on the 1-shot 5-way OMNIGLOT dataset with varying router depths

However, combining many routing layers might improve performance, such as in the case of MRN-GB. We will further investigate this behavior further in future work.