# OpenReview forum: "Efficient Heterogeneous Meta-Learning via Channel Shuffling Modulation"
_ICLR.cc/2024/Conference — ICLR 2024 poster_

### Official Review · Reviewer_KAtC · 2023-10-30

**Soundness:** 3 good
**Presentation:** 3 good
**Contribution:** 3 good
**Rating:** 6
**Confidence:** 5

**Summary:**

The paper addresses the issue of meta-learning methods assuming a uniform task distribution, which hinders their performance on multi-modal task distributions. To address this, the authors introduce a new strategy for heterogeneous meta-learning that modulates the routing between convolution channels in neural networks. This strategy, called the Gumbel-Benes layer, offers a more efficient parameter complexity compared to existing methods and shows improved performance in terms of accuracy and runtime on multi-modal meta-learning benchmarks.

**Strengths:**

**Originality**: The paper innovatively tackles the homogeneous task distribution assumption in meta-learning by introducing a modulation mechanism between convolution channels, showcasing a fresh approach to the problem.

**Quality**: The proposed Gumbel-Benes layer is robustly validated on multiple multi-modal meta-learning benchmarks, evidencing its effectiveness and superiority over existing methods.

**Clarity**: The presentation of the novel strategy and its underlying concepts is coherent, making the methodology and results easily understandable.

**Significance**: By addressing the limitations of previous meta-learning methods and offering a more efficient parameter complexity solution, this work holds substantial potential to influence future research and applications in the field of meta-learning.

**Weaknesses:**

Overall, the paper exhibits commendable novelty, particularly in utilizing meta-routing to learn task-specific activations for diverse tasks. However, I have several concerns related to the experiments:

1. What is the impact of the lambda in front of the prototype loss on the model's performance?
2. Did the authors experiment with adding routers only in shallow or deep layers of the network? I'm curious about the influence of routers on the model across different layers.
3. Concerning the complexity introduced by the Router to the model, could the authors provide comparisons in terms of parameter count and training efficiency with HSML?

**Questions:**

1. Given that $\phi(D_1)$  produces the prototype for the entire task rather than for individual classes, I'm curious: is $L_{PROTO}$'s loss computed for each class's prototype?

2. A more in-depth theoretical analysis would undoubtedly solidify the paper further and provide a more comprehensive view.

3. It would greatly benefit readers if the authors could include an algorithm in the appendix, offering clearer insight into the paper's approach.

---

> ### Author Response · Authors · 2023-11-20
> **Thank you for your comments**
>
> 1. We provide below the results of varying the $\lambda$ parameter in the MRN-GB/MRN-GS objective function. We used the 1-shot 5-way Omniglot dataset and 200 training epochs for each experiment (Note that in the full experiment in our manuscript we used 2000 training epochs, hence the accuracy values reported below are somewhat lower).
>
> | $\lambda$         | 0.2   | 0.5   | 1.0       | 2.0       | 5.0   |
> |-------------------|-------|-------|-----------|-----------|-------|
> | MRN-GB (Omniglot) | 0.827 | 0.829 | **0.888** | 0.861     | 0.840 |
> | MRN-GS (Omniglot) | 0.843 | 0.848 | 0.861     | **0.885** | 0.867 |
>
> The above result seems to suggest that the parameter $\lambda$ cannot be too large, causing the loss function to behave similarly to the ProtoNet loss; or too small, causing the task embedding module to capture unmeaningful patterns.
>
> 2.  In all of our experiments, we used a base architecture with 4 convolution blocks. For our methods, we added one routing layer after every block, hence the routers are distributed at various depths of the network. The influence of the routers at each depth is an interesting experiment to conduct. We have provided some preliminary results below (also on the 5-way 1-shot Omniglot dataset), in which we compared the performance of our methods using all 4 routers vs. the performance of using only one router at different depths (e.g., larger depth means being closer to the output layer).
>
> | Router Depth      | 1     | 2     | 3     | 4     | All   |
> |-------------------|-------|-------|-------|-------|-------|
> | MRN-GB (Omniglot) | 0.873 | 0.875 | 0.869 | 0.747 | 0.888 |
> | MRN-GS (Omniglot) | 0.851 | 0.867 | 0.835 | 0.805 | 0.861 |
>
> Our result suggests that having the routing layer placed earlier on in the architecture is the most effective practice (e.g., putting a routing layer after the second convolution block gives the best performance). However, combining many routing layers might improve performance, such as in the case of MRN-GB. We will investigate this behavior further in future work.
>
> 3. Parameter count and training efficiency comparison have been provided in Table 3 of our manuscript.
>
> 4. The ProtoNet loss ($\mathcal{L}_{proto}$) is computed for each class in a meta task. The embedding vector phi is the **average** of the ProtoNet embeddings for all classes in the task. This is a well-known strategy to compute task embedding. Thank you for pointing it out, we will add an algorithm box to clarify this in our revision.

---

### Official Review · Reviewer_d54Q · 2023-10-31

**Soundness:** 3 good
**Presentation:** 3 good
**Contribution:** 3 good
**Rating:** 6
**Confidence:** 4

**Summary:**

This paper introduces a new meta-learning approach for heterogeneous sets of tasks. The method is presented as a permutation learning problem in which convolution channels are shuffled via a continuous switch network. This trainable routing network is parameterized by input embeddings learned via the prototypical loss (Snell et al., 2017), in which the centroids encode the prototype shuffling parameters corresponding to each task. The method is favorable compared to using Gumbell-Sinkhorn layers (Mena et al., 2018) which use routing embedding dimensions quadratic in the number of channels, hence requiring larger networks with more parameters and longer inference times. The proposed method is also empirically compared to a few other baseline algorithms that do not account for heterogeneity, matching SOTA performance in single task regime and outperforming alternatives in the heterogeneous task setup.

**Strengths:**

Originality and significance: The presented method shows clear improvements over the studied baselines by showing how a task-based ordering can be derived from a smaller dimensional embeddings: $C \log C$ v.s. $C^2$. The empirical analysis uses both problems with synthetically generated tasks (JIGSAW), as well as problems formulated by grafting multiple tasks. The results show the effectiveness in heterogeneous meta-learing domains.

Quality and clarity: The paper is easy to follow and the experiments are explained in sufficient detail.

**Weaknesses:**

The paper is not self-contained and requires prior knowledge of prototypical networks and Gumbel-Sinkhorn layers. A reader not familiar with both topics will have to consult the references for basic definitions of key components of the proposed method (namely $L_{proto}$ and  $GS(.)$). A minimal overview of these should have been included in the main body of the paper, or at least in the appendix.

**Questions:**

Even though the method reduces the nominal parameter complexity of the required routing network, it might still be over parametrizing the space by a log factor. Specifically, in order to define an ordering of $C$ channels one should require no more than $C$ values (each with at least $\log_2C$ bits). I suspect that one can derive a (continuous) shuffling using embeddings of size $C$ instead of $C \log C$ by using a sorting network (e.g. Bitonic sort) in a similar fashion as the Benes network is used in this work, but with a different switch type: $(U_j, channel_j)$ pairs are sorted by the key values and deltas between key values (passed a sigmoid) can form continuous switches. Can such a mechanism work in the use cases presented by the paper, and how would it compare to using the MRM-GB in terms of parametrization and evaluation speed?

---

> ### Author Response · Authors · 2023-11-20
> **Thank you for your comments**
>
> 1. We will provide an overview of related concepts in the revision of our paper.
>
> 2. Thank you for the interesting suggestion regarding the use of sorting networks. It is true that other sorting networks can be used in place of the Benes routing network for our strategy. This approach would be very similar in spirit to our method. Whereas our Gumbel Benes layer is conditioned on the task embedding vector, a sorting layer would theoretically derive contextual information from learning a value function that scores the convolution channels. Furthermore, a bitonic sorting network will also require $C\log^2 C$ comparators, so we would expect this mechanism to have, at best, similar complexity to our method.

---

> > ### Comment · Reviewer_d54Q · 2023-11-20
> > **Updates**
> >
> > 1. IIUC no revision was uploaded for this submission that adds missing parts mentioned in my review.
> > 2. Note that even though the complexity of the sorting network is the same as Benes, its parameter count is smaller by a log factor. Hence the model that generates the routing parameters can have lower output dimension and hence lower complexity.

---

> > > ### Author Response · Authors · 2023-11-21
> > > **Discussion**
> > >
> > > 1. Thank you for reminding us. We have provided the revision as requested.
> > > 2. Thank you for explaining. We assume that you might be talking about the reference "Differentiable Sorting Networks for Scalable Sorting and Ranking Supervision" (Petersen et al., 2021). We now understand and agree that, in the simplest form, each sorting layer needs only maintain a C-dimensional vector to encode a shuffling of C channels. However, in practice, we suppose that extra parameterization would be required to meaningfully learn this vector from data. Nonetheless, this is definitely an interesting alternative that we will try to implement and compare against in future revisions.

---

### Official Review · Reviewer_aRVA · 2023-10-31

**Soundness:** 2 fair
**Presentation:** 3 good
**Contribution:** 3 good
**Rating:** 6
**Confidence:** 5

**Summary:**

**Edit: I have raised my score to 6 to reflect the author's updates to the submission.**


The paper considers the problem of meta-learning across heterogeneous tasks when the model is expected to extract and generalize the meta-knowledge and transfer it to quickly learn the novel tasks. Whereas, many meta-learning models focus on the tasks coming from the same distribution (homogeneous), the authors consider the directly the tasks coming from a multi-modal task distribution (heterogeneous setting). Recently, many appeared many methods that tackle the heterogeneous meta-learning, but they suffer from a significant increase in parameter complexity. As an alternative approach, the authors propose a novel strategy which incorporates modulating the routing between convolution channels in the network. The paper introduce a novel neural neural permutation layer based on the classical Benes routing network. Finally, the proposed method is compared against various meta-learning benchmarks.

**Strengths:**

The paper has a few significant strengths overall, which I will outline below:
1. Considering directly the heterogeneous tasks setting which I found especially significant within the Meta-Learning.
2. The main idea of introducing the Gumbel-Benes routing layer is very interesting and seems to be novel in Meta-Learning.
3. Leveraging the complexity of permutation layer inspired by the Benes network is fair.
4. I like experiment presented in Figure 3.
5. The presentation is clear. Overall, the flow of the manuscript is well-organized.

**Weaknesses:**

However, despite the strengths, the paper has a few major and minor weaknesses. I will focus on the experiments section especially, because I recognize it as insufficient:

1. The results of all methods, presented in the Table 1 and Table 2, are usually within their standard deviations. Because of that, I will not support the claim that the presented method is significantly better than others.
2. The methods that are chosen for the comparison are not a current state-of-the-art methods, having a few years. I strongly encourage authors to include comparison with other methods like [1] or [2].
3. The comparison are considered only across a single neural network architecture. I admit that the scenarios of using the Gumbel-Benes routing layers are limited into CNN architectures. However, there is not the reason to skip the comparison on highly popular in Meta-Learning, ResNet architectures. I will suggest for sure comparison on at least ResNet-10 (as in, e.g., [3]).
4. The lack of more challenging 1-shot 5-way classification on Mini-Imagenet dataset.
5. As far as I know, there is the first example of the Benes networks-inspired Meta-Learning approach. However, there were others works based on the Benes routing, e.g, [4]. I would like to see the Related Work section given credits to that works and comparing the proposed method with them.

**References:**

[1] Sun, Z., Wu, J., Li, X., Yang, W., and Xue, J.-H. Amortized bayesian prototype meta-learning: A new probabilistic meta-learning approach to few-shot image classification. In International Conference on Artificial Intelligence and Statistics, pp. 1414–1422. PMLR, 2021.

[2] Ye, H.-J., Hu, H., Zhan, D.-C., Sha, F.: Few-shot learning via embedding adaptation with set-to-set functions. In: IEEE/CVF Conference on Computer Vision and Pattern Recognition (CVPR), pp. 8808–8817 (2020)

[3] Patacchiola, M., Turner, J., Crowley, E. J., O’Boyle, M., and Storkey, A. J. Bayesian meta-learning for the few-shot setting via deep kernels. Advances in Neural Information Processing Systems, 33, 2020.

[4] Freivalds, K., Ozoliņš, E., & Šostaks, A. (2019). Neural shuffle-exchange networks-sequence processing in o (n log n) time. Advances in Neural Information Processing Systems, 32.

**Questions:**

I would like to see especially the following experiments and improvements:
1. Provide results for the 1-shot 5-way classification setting on Mini-Imagenet.
2. Please, compare with state-of-the-art methods as suggested in Weaknesses section.
3. The work will be better if you could test the Gumbel-Benes layers in ResNets also.
4. Add the Related Work section regarding other deep learning works utilizing the Benes networks/routing algorithm.

**Questions:**

1. Another place where the Gumbel-Benes routing layers might be beneficial are standard methods working on the Meta-Dataset [1], which is a created really heterogeneous tasks dataset. The best methods working on Meta-Dataset are often ResNets with some adapting layers within, so they are really within the scope of the Gumbel-Benes layers. For the example methods, please see the rank tables in [2].


**References:**

[1] Triantafillou, E., Zhu, T., Dumoulin, V., Lamblin, P., Evci, U., Xu, K., ... & Larochelle, H. (2019). Meta-dataset: A dataset of datasets for learning to learn from few examples. arXiv preprint arXiv:1903.03096.

[2] https://github.com/google-research/meta-dataset

---

> ### Author Response · Authors · 2023-11-20
> **Thank you for your comments**
>
> 1. Regarding significance, even though the margin of improvement is not as large as you would prefer, we would like to highlight that our contribution also includes a parameter efficient meta learning formulation, as demonstrated in Table 3.
>
> 2. Following your suggestion, we have conducted follow up experiments on the more difficult 1-shot, 5-way Mini-ImageNet setting and its Jigsaw counterpart. We adopt similar training settings to the 5-shot, 5-way experiments in the main manuscript. That is, we trained every baseline for 2000 epochs, sampling 32 training tasks of 16 images per epoch (1-shot, 15-queries). For testing, we sampled 5 test tasks with 1 training and 15 test images per task. Due to the limited rebuttal time, we only focus on comparing with other heterogeneous Meta learning baselines (i.e., MMAML, HSMAML, URL). We have also provided the result using ResNet-18 on these scenarios to answer your other question. The results of these experiments are detailed in the Table below:
>
> | Dataset/Method                   | MRN-GB    | MRN-GS | MMAML | HSML      | URL   |
> |----------------------------------|-----------|--------|-------|-----------|-------|
> | Mini-ImageNet                    | **0.504** | 0.496  | 0.432 | 0.472     | 0.496 |
> | Jigsaw Mini-ImageNet             | **0.480** | 0.464  | 0.432 | 0.408     | 0.448 |
> | Mini-ImageNet (ResNet-18)        | 0.547     | 0.526  | 0.541 | **0.555** | 0.470 |
> | Jigsaw Mini-ImageNet (ResNet-18) | **0.525** | 0.299  | 0.490 | 0.448     | 0.501 |
>
> 3. We have compared our method to a lot of benchmarks in the manuscript, one of which (i.e. the URL method) was published very recently in 2022. Due to the limited rebuttal time, we may not be able to provide results to the suggested methods. In particular, we couldn’t find an implementation of reference [1]. On the other hand, FEAT (reference [2]) adopts a Transformer architecture that does not immediately fit with our permutation framework, thus it would require significant changes to provide an apples-to-apples comparison. That being said, we will try our best to provide such results in our future revision.
>
> 4. Thank you for your suggestion. We will provide a discussion of other DL works that use the Benes Network in our revision. We however hope you agree that our work is the first that uses this concept in a meta-learning context.

---

> > ### Comment · Reviewer_aRVA · 2023-11-20
> >
> > I want to thank the authors for their response and including additional results. In the following paragraphs, I will go through the points from the authors response to my review.
> >
> >
> > 1. Thank for this clarification. However, I have two concerns regarding this point:
> >
> >
> > - The presented results don’t compare with the methods having better results - e.g., in response to my review, the authors provide the comparison on mini-ImageNet 5-way 1-shot task, achieving 50.4% of accuracy while even in [1] (from my review, see Table 1), we have at least 5 methods achieving better accuracy (using the same backbone).
> >
> >
> > - Regarding the results from Table 3 - I agree that the authors need smaller number of parameters and get lower runtime than other presented methods. However, my main concern is that the comparison is unfair. The authors proposed to use 4-layers CNN networks and increase the number of channels, which is strongly favorable for their method. However, the most common setting for, e.g., MAML, is to use different backbone - usually some deeper ResNet to get the better results. Because of that, the usual runtimes are not that big as provided (since other methods rather not use such architectures). From my perspective, the comparison of the methods in Table 3 is provided on artificial task, being far from standard setting.
> >
> >
> > 2. Thank for providing the additional results. I assume that including them into manuscript would make the work better. Regarding results, please look my comment above.
> >
> >
> > 3. I will be looking forward to see additional baselines in the revised version of the manuscript, as promised by the authors.
> >
> >
> > 4. Thank you for promising of adding the discussion in the next version of the manuscript. I agree with the authors that, as far as I know, there is the first usage of this concept in the context of meta-learning.
> >
> >
> > Overall, I think that the authors are solving unrealistic setting - i.e., their experiments on heterogeneous meta-learning tasks are usually not a problem for many meta-learning approaches (e.g., the methods compared on Meta-Dataset). I’m also concern with the lack of comparison with the methods achieving better accuracy - you can see, e.g., methods presented in Tables as baselines in [1] (from my review).
> >
> >
> > Finally, I thank the authors for the response and including the additional results. I agree that the proposed method is interesting. However, I think that my concerns were not addressed properly and lack of including other baselines might be confusing for many readers coming from the few-shot learning community and knowing the typical baselines results on the presented tasks. Unfortunately, in this situation, I just cannot raise my score.

---

> > > ### Author Response · Authors · 2023-11-20
> > > **Thanks for your prompt responses**
> > >
> > > 1. Regarding your concern about not matching the performances of state-of-the-art methods, we would like to refer you to appendix 4 of reference [1], which says that their mini-ImageNet experiments used 12000 training epochs, whereas our only used 2000 training epochs. With 1/6 the training budget, it would not be fair to directly compare our results to Table 1 of reference [1]. If you would accept the quoted performances from [1], we would gladly rerun our experiments for 12000 training epochs for a fairer comparison.
> > >
> > > 2. Regarding the ResNet backbone, we have provided the extra results in the rebuttal. We also would like to highlight that the 4-layer CNN architecture is a standard backbone provided by the L2L package (Arnold et al., 2020, "learn2learn: A Library for Meta-Learning Research") and was by no mean chosen to favor our method. In fact, the ResNet backbone is **not** the most common setting for MAML, although it might have been used in some meta learning papers. In the original MAML paper itself, I believe the same architecture with 4 conv blocks was used (see page 6 of Finn et al., 2017).
> > >
> > > 3. Regarding the concern about unrealistic setting, we believe most of our experiments do align with previous heterogeneous meta-learning/ few-shot learning literature. For example, the jigsaw puzzle type of tasks was introduced in (Su et al., 2020, "when does self-supervision improve few-shot learning?"), whereas the dataset grafting task (with the exact dataset combination) was introduced in (Yao et al., 2019, "Hierarchically Structured Meta-learning"). Thus, we confidently say that we do not invent anything new to favor our method.
> > >
> > > 4. Regarding the lack of baselines, there are now hundreds of meta learning methods, and it would be impossible to compare to all. With all due respect, we think that pointing out baselines that one does not compare to is hardly constructive. We have no doubt that the references you pointed out are strong baselines, but can you please explain why our selected baselines are no longer relevant. In fact, the URL baseline (Li, Liu and Bilen, 2021, "Universal Representation Learning from Multiple Domains
> > > for Few-shot Classification") is both well-cited and more recent than all methods presented in Table 1 of reference [1]. Both MMAML (Vuorio et al., 2019) and HSML (Yao et al., 2020) are also very well-cited, and definitely not older than most methods presented in Table 1 of reference [1].

---

> ### Author Response · Authors · 2023-11-21
> **Extra results of our methods trained for 12000 epochs to compare with Table 1 of Reference [1]**
>
> Dear reviewer,
>
> We have re-run our method for 12000 training epochs. The table below presents our results side by side with the results quoted from Table 1 of Ref [1].
>
> | Method                                      | MRN-GB (CNN) | MRN-GS (CNN) | BMAML | PLATIPUS | ABML | Amortized VI | VERSA | Meta Mixture | VAMPIRE | DKT   | ABPML (Ref [1]) |
> |---------------------------------------------|--------------|--------------|-------|----------|-------|--------------|-------|--------------|---------|-------|-----------------|
> | Mini-ImageNet (1-shot, 5-way, 12000 epochs) | **0.606**       | 0.591        | 0.538 | 0.501    | 0.450 | 0.441        | 0.534 | 0.512        | 0.515   | 0.497 | 0.533           |
>
> As you can see, given similar training budget our method is capable of achieving much better performance than the quoted results. We hope this would give you sufficient evidence to support our method's potential for now. Given time, we will thoroughly compare with these previous methods on the exact same setting (train/test split, learning rate, layer size, etc.)

---

> > ### Comment · Reviewer_aRVA · 2023-11-22
> >
> > I would like to thank the authors for their response for my concerns. In the following paragraphs, I will go through all the issues mentioned by authors in the last two responses.
> >
> >
> > 1. Many thanks for adding the number of the training epochs between the methods and for the rerunning their code also for the full 12k epochs. I will go to the achieved results later in the response.
> >
> >
> > 2. Regarding the ResNet and Conv-4 backbones. I totally agree that they are both used in the few-shot learning papers. However, I just want to mention that my comment regarding the used backbone in Table 3 was not intended to criticize the Conv-4 backbone in the few-shot learning settings, but was to emphasize the artificially of this comparison. If it was not crystal clear, I will add explanation. I think that the Table 3 looks good up to the 64 number of channels. Why I think that? Since in the Meta-Learning architecture the most popular backbone is the Conv-4 with the 64 channels in each layer. This particular backbone with larger number of channels per layer is being seldom used just because it doesn’t scale well (what we can see in the last two rows of Table 3). In practice, for the harder tasks, demanding the more powerful backbone, the few-shot methods usually go for the ResNets. Now, my concerns about the Table 3 was the following. For a reader of this paper, the results now presented in Table 3 might suggest that for harder tasks, the MAML-based methods just cannot be used in practice, because they barely fit in the GPU. Whereas, in practice, the different backbone (popular for more demanding tasks) might be successfully used with different backbone. I want also to apologize the authors if my concerns were not crystal clear at the beginning and I hope that they are better understandable now.
> >
> >
> > 3. I agree with the authors that the presented datasets were previously known in the few-shot learning field. My concern was that comparing to the mentioned, e.g., Meta-Dataset or Meta-Dataset+VTAB (which also contains most of the datasets from the paper) the experimental setting seems to be reasonably easier. That’s why I would prefer experiments on the mentioned datasets. Moreover, for the future, I strongly encourage the authors to try their method + e.g., Transformer/ResNet backbone on Meta-Dataset, because I suppose that it might be interesting and possibly beneficial for the field.
> >
> >
> > 4. Regarding lack of baselines, I’ve just believe that presenting the baselines that are significantly worse than the current SOTA, might provide some misunderstanding for the readers. That’s why I mentioned just a few stronger baselines to show the proposed method in wider context.
> >
> >
> > 5. I would like to congratulate the authors on the outstanding improvement in results presenting on miniImageNet 1-shot 5-ways task. I hope that in the revised version of the manuscript, the authors will provide the reruned experiments to show what are the real practical limits of their method. Just for the quick test could the authors check their method rerun for 12k epochs for the Omniglot task from Table 1 to show if the method can do significantly better than MAML (I’ve selected the easiest dataset because running time of the discussion)? Regarding the presented new results on the miniImageNet and last response of the authors, I am still a little bit concerned about the fairness of comparison with other methods. In the last response, the authors wrote that: "Given time, we will thoroughly compare with these previous methods on the exact same setting (train/test split, learning rate, layer size, etc.)", which I don’t know how to exactly understand. I want to ask what backbone architecture was used in this particular setting? I assume it was CNN but could the authors explain how many layers and channels were used there? I’m curious what the "exact same setting" means here since all of these methods were compared on exactly same setting - e.g., exactly same CNN backbone. I hope that the authors could explain it to me and dispel my last concerns.

---

> ### Author Response · Authors · 2023-11-22
> **Thank you for your response**
>
> First of all, thank you for clarifying your points. We do understand your concerns better now, and appreciate your constructive feedback. To add some discussion:
>
> We acknowledge that ConvNet layers with more than 64 channels are not as practical in practice, and there are better architectures to try our method on. Some, such as ResNet, are immediately compatible as shown in the rebuttal experiments and achieve reasonably good performance. We will heed your advice and experiment with Transformer/ViT architectures in the future, possibly by permuting the multi-head structure of transformer. We would also like to highlight that the heterogeneous meta-learning baselines that adopt the weight modulation strategy will become more impractical with increasing model size, regardless of the architecture type. In our manuscript we chose to increase the number of channels to achieve this effect. Although it was not meant to favor our method, we do understand how confusion can arise.
>
> Regarding the rerun experiments on 12k epochs, the backbone that we used is the CNN architecture with 4 conv blocks and 4 routing layers as described in the paper. Each conv block has a 64 channel conv layer, followed by BatchNorm, ReLU and MaxPool, which is exactly similar to [1]. What we meant in our previous response was that we couldn't find their implementation, so we cannot reproduce exactly their learning rate, train/test split, weight initialization (our apology, we meant to say layer weight, not layer size). Reference [1] also did not describe how many test images are there in one test task, and how many query images are there in one training task. In our experiments, each task consists of 1-shot/15-query or 1-shot/15-test. Due to limited time, we also did not go into each method listed in Table 1 of reference [1] to check their experimental setting to make sure our results are as fair as possible.
>
> Finally, we will try our best to provide the accuracy of our methods on the Omniglot dataset with 12k epochs in the remaining time of the rebuttal.

---

> > ### Comment · Reviewer_aRVA · 2023-11-22
> >
> > Dear authors!
> >
> >
> > Thank you for your response and better understanding to my concerns. I strongly encourage you to include my suggestions stated in the previous responses in the revised version of the manuscript. I suppose that including parts of our discussion (regarding backbone architectures) as an ablation study might be beneficial for the paper.
> >
> >
> > In this situation, I would like to increase my score to 6.

---

> > > ### Author Response · Authors · 2023-11-23
> > > **Thank you for the valuable discussion and fair judgement**
> > >
> > > To wrap up our discussion, we would like to provide the performance of our method on the Omniglot dataset over 12k epochs.
> > >
> > > | Method                                      | MRN-GB (CNN) | MRN-GS (CNN) | BMAML | PLATIPUS | ABML | Amortized VI | VERSA | Meta Mixture | VAMPIRE | DKT   | ABPML (Ref [1]) |
> > > |---------------------------------------------|--------------|--------------|-------|----------|-------|--------------|-------|--------------|---------|-------|-----------------|
> > > | Omniglot (1-shot, 5-way, 12000 epochs) |   0.991    | 0.989 | - | -    | - | 0.978        | **0.997** | -        | 0.984   | - | 0.988           |
> > >
> > > Again, the number for the other methods are quoted directly from Table 1 in reference [1]. Although we did not achieve the absolute best performance here, I think it's fair to say that all baselines have more or less perfected this relatively easy task, and the margin between methods are quite insignificant.
> > >
> > > We have also incorporated all the rebuttal numbers into a new revision as you suggested.
> > >
> > > Best regards,
> > > Authors

---

> > > > ### Comment · Reviewer_aRVA · 2023-11-23
> > > >
> > > > I would like to thank the authors for adding the additional results in the very limited time before the end of discussion. It's good to see that the proposed method achieves better results than presented in the first version of the paper. I suggested this experiment on the easy task just to see if there might be any consistency between additional training time and achieving better results than the ones presented at the beginning.
> > > > Once again, I hope to see in the final version the additional experiments and ablation studies that were suggested during our discussion.
> > > >
> > > > Best regards,
> > > > Reviewer aRVA

---

### Official Review · Reviewer_ppvH · 2023-11-04

**Soundness:** 3 good
**Presentation:** 3 good
**Contribution:** 3 good
**Rating:** 6
**Confidence:** 4

**Summary:**

This paper deals with the heterogeneous meta-learning problem from the perspective of channel shuffling modulation.
Instead of the conventional fixed backbone + modulation layer scheme for multiple task distributions, this paper seeks to find a task-specific channel permutation routing mechanism.

The idea is motivated by ShuffleNet and implemented with a task-specific prototypical vector to learn the shuffling operation. For the permutation matrix, this paper first adopts the Gumbel-Sinkhorn layer to generate the permutation. Then, the classical Benesˇ routing network is utilized to improve the efficiency.

Experiments are conducted on several heterogeneous datasets and compared with various MAML-based methods. As a parameter-efficient method, the proposed method also performed generally better than baselines.

**Strengths:**

- (1) This paper provides a different perspective for heterogeneous meta-learning, i.e., channel shuffling by permutation learning. It can be a good alternative to the modulation-based methods.
- (2) The utilization of the traditional Beneˇs network can improve the efficiency of the proposed method.
- (3) Experiments are conducted on the meta-learning benchmark, and the results are fair and generally better than other MAML-variant methods.

**Weaknesses:**

- (1) The authors describe the heterogeneous meta-learning problem from the MAML perspective. However, similar to heterogeneous meta-learning, researchers also used the terminology multi-domain/cross-domain to describe the diverse task distribution when dealing with few-shot meta-learning problems, e.g., [R1]. Besides Li et al. (2022), other related works applying feature modulation can also be discussed, such as [R2, R3].
- (2) I am wondering why permutation and channel shuffling work compared with modulation. The intuition of modulation assumes a strong shared backbone, and each dataset/distribution performs task-specific adjustments for its data. For the permutation case, it is better to visualize what permutation can be learned for different distributions.


[R1] Triantafillou, E., Zhu, T., Dumoulin, V., Lamblin, P., Evci, U., Xu, K., Goroshin, R., Gelada, C., Swersky, K., Manzagol, P.A. and Larochelle, H., 2019. Meta-dataset: A dataset of datasets for learning to learn from a few examples. arXiv preprint arXiv:1903.03096.

[R2] Triantafillou, E., Larochelle, H., Zemel, R., & Dumoulin, V. (2021, July). Learning a universal template for few-shot dataset generalization. In International Conference on Machine Learning (pp. 10424-10433). PMLR.

[R3] Liu, Y., Lee, J., Zhu, L., Chen, L., Shi, H., & Yang, Y. (2021). A multi-mode modulator for multi-domain few-shot classification. In Proceedings of the IEEE/CVF International Conference on Computer Vision (pp. 8453-8462).

**Questions:**

Please see above.

---

> ### Author Response · Authors · 2023-11-20
> **Thank you for your comments**
>
> 1. Thank you for your suggestion; we will provide a discussion of the suggested related works in our revision.
>
> 2. We strongly agree that the intuition of modulation assumes a strong shared backbone, and each dataset/distribution performs task-specific adjustments for its data. However, the degree of adjustments will depend on the heterogeneity of the task distribution. Additive modulations might suffice for task distributions consisting of closely related tasks. On the other hand, more challenging distribution of tasks tend to require more significant modifications, as explored in previous studies of heterogeneous meta learning via modulation. Our hypothesis is that structural modulations, such as a shuffling of the embedded information channels, would be more effective at modulating between tasks of different modalities. In fact, this is not an arbitrary design choice as it is also supported by previous empirical evidence. For example, random shuffling has been used in the context of compact neural architecture due to its efficiency in encoding information.
>
> 3.  It would be possible to visualize the learned permutations. However, we would like to note that the identity of the permutations themselves are not quite important. We believe that it is more critical to show that the network can learn to assign similar tasks to similar permutations, and different tasks to different permutations, and thus improve knowledge organization. To show this, we have provided the t-SNE plots of the learned Benes switch configurations for 1000 tasks in two different scenarios (see our Appendix A), which show distinct clusters that correspond to the task identities.

---

> > ### Comment · Reviewer_ppvH · 2023-11-23
> >
> > Thanks for the response. My concerns are addressed.
> > I will keep my Rating as 6.

---

### Author Response · Authors · 2023-11-21
**Revised manuscript**

Dear reviewers,

We have uploaded a revision of our manuscript as promised. The changes include Appendix A, D and text highlighted in red in the main paper.

To address the concern of reviewer ppvH, we have incorporated references [R2, R3] into our related work section. To be more explicit, we would like to clarify that [R2] adopts the adapter strategy, similar to URL (Li et al., 2021), whereas [R3] adopts the weight modulation strategy, similar to MMAML (Yao et al., 2019) and HSML (Vuorio et al., 2019). Thus, the same positioning argument in our related work section will still apply.

To address the concern of reviewer aRVA, we have incorporated references [1, 3] and [4] into our related work section. We acknowledge that references [1, 3] are valid baselines and will soon provide our results using a similar experimental setting to fairly compare with the quoted results in [1]. We have credited reference [4] (thanks for pointing it out), but we would like to highlight to all reviewers that it is not dealing with the meta-learning setting, nor is our technique similar to the technique in [4].

To address the concern of reviewer d54Q, we have added Appendix D to explain related concepts such as ProtoNet loss and Gumbel-Sinkhorn.

To address the concern of reviewer KAtC, we have added Appendix A which details our training algorithm.

Once again, we thank the reviewers for your constructive feedback. We would like to hear back from you and further incorporate your valuable comments into our manuscript.

---

### Meta-Review · Area_Chair_debi · 2023-12-07

**Metareview:**

Original concerns raised by the reviewers include 1, the difference between the proposed method and related work in terms of methodology is unclear, experimental setups are not convincing, comparison results with SOTA baselines are missing, etc. After rebuttal, most of the concerns have been addressed. Therefore,  based on the new shape of this paper, I recommend an acceptance.

**Justification For Why Not Higher Score:**

This paper is above the acceptance threshold, but not good enough for a spotlight presentation.

**Justification For Why Not Lower Score:**

This work deserves to be published.

---

### Decision · Program_Chairs · 2024-01-16

Accept (poster)